# Do Deep Reinforcement Learning Agents Model Intentions? †

**Tambet Matiisen \***, **Aqeel Labash**, **Daniel Majoral, Jaan Aru and Raul Vicente**

Institute of Computer Science, University of Tartu, Narva mnt 18, 51009 Tartu, Estonia
* Correspondence: tambet.matiisen@ut.ee
† This paper is an extended version of our paper published in Adaptive Learning Agents 2018 workshop.

**Abstract:** Inferring other agents' mental states, such as their knowledge, beliefs and intentions, is thought to be essential for effective interactions with other agents. Recently, multi-agent systems trained via deep reinforcement learning have been shown to succeed in solving various tasks. Still, how each agent models or represents other agents in their environment remains unclear. In this work, we test whether deep reinforcement learning agents trained with the multi-agent deep deterministic policy gradient (MADDPG) algorithm explicitly represent other agents' intentions (their specific aims or plans) during a task in which the agents have to coordinate the covering of different spots in a 2D environment. In particular, we tracked over time the performance of a linear decoder trained to predict the final targets of all agents from the hidden-layer activations of each agent's neural network controller. We observed that the hidden layers of agents represented explicit information about other agents' intentions, i.e., the target landmark the other agent ended up covering. We also performed a series of experiments in which some agents were replaced by others with fixed targets to test the levels of generalization of the trained agents. We noticed that during the training phase, the agents developed a preference for each landmark, which hindered generalization. To alleviate the above problem, we evaluated simple changes to the MADDPG training algorithm which lead to better generalization against unseen agents. Our method for confirming intention modeling in deep learning agents is simple to implement and can be used to improve the generalization of multi-agent systems in fields such as robotics, autonomous vehicles and smart cities.

**Keywords:** multi-agent reinforcement learning; theory of mind; artificial neural networks





## 1. Introduction

The ability of humans to infer the mental states of others, such as their beliefs, desires, or intentions, is called theory of mind (ToM) [1,2]. Inferring other agents' intentions gives an advantage both in cooperative tasks, where participants have to coordinate their activities, and in competitive tasks, where one might want to guess the next move of one's opponent. Predicting other agents' unobservable intentions from a few observable actions has important practical applications. For example, with self-driving cars, the behavior modeling of other traffic participants is seen as a crucial ingredient of human-level driving capability [3,4].

In this work, we investigate to which degree artificial agents trained with the multi-agent deep deterministic policy gradient (MADDPG) deep reinforcement learning algorithm [5] have the ability to infer the intentions of other agents. Our experiments were based on a cooperative navigation task from [5], where three agents have to cover three landmarks and coordinate between themselves to decide which agent covers which landmark (see Figure 1). We applied a linear readout probe to each agent's hidden-layer activations and tried to predict the final landmarks covered by other agents at the end of the episode. If early in the episode, the other agents' final landmarks can be accurately predicted, then we could claim that the agents are representing information about others' specific plans, and hence to some extent infer their intentions.

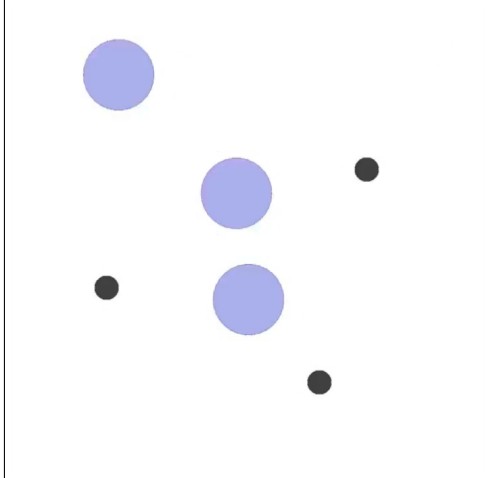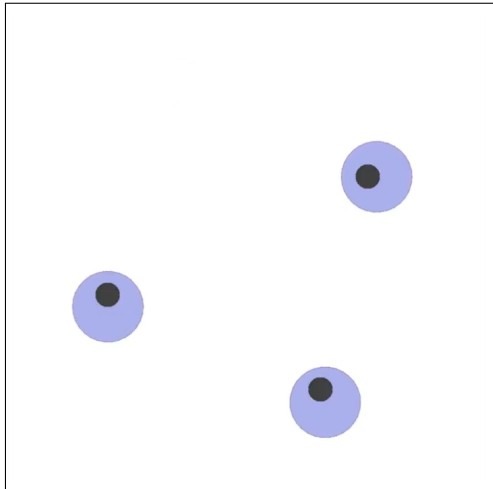

**Figure 1.** Cooperative navigation task. Large purple circles are agents; small black circles are landmarks. The reward scheme incentivizes the coverage of each landmark by a different agent without collisions. (**Left**): Initial layout, agents and landmarks are positioned randomly. (**Right**): Final result; each landmark is covered by one agent.

In our experiments, we indeed showed that the intentions of other agents can be decoded from an agent's hidden-layer activations using a linear decoder, though the same information cannot be linearly decoded from the observations only (based on which, the hidden-layer activations are computed). This means that agents apply a learned transformation to the observation, which makes this information more explicit in hidden layer 1. Interestingly, the same information can be decoded less accurately from hidden layer 2.

Probing the agents' minds to read out their intentions clearly showed that training multiple agents jointly using reinforcement learning leads to severe co-adaptation where agents overfit to their training partners and do not generalize well to novel agents. For example, we demonstrate that in a cooperative navigation task, the agents trained using the MADDPG algorithm develop favorite landmarks, and these are different for all three trained agents. The lack of generalization is exposed when a MADDPG-trained agent is put together with two "Sheldon" agents—agents that always go to the same fixed landmark (named after a character in "The Big Bang Theory" who insisted on sitting in the same spot on the couch.). We show that the performance of an agent degrades substantially when the only remaining available landmark is its least-favorite one.

Finally, we evaluate a number of changes to the MADDPG algorithm that make this problem less severe. In particular, randomizing the order of agents for each episode improves the generalization result, whereas the ensembling suggested in [5] does not.

The main contributions of the paper are:

- We show that deep reinforcement learning agents trained with the MADDPG algorithm in cooperative settings learn models which represent information about the intentions of other agents.
- We show that jointly trained MADDPG agents co-adapt to each other and do not generalize well when deployed with agents that use unseen policies.
- We evaluate a number of changes to the MADDPG algorithm that alleviate the generalization gap to some degree.

## 2. Related Work

There is a wealth of literature about modeling other agents in multi-agent systems using classical methods [6], but we specifically focus on the use of deep neural networks for the task [2]. Furthermore, while most of the work in the area of ToM for artificial agents focuses on evaluating their beliefs—for example, if they pass some variation of the

Sally-Anne test [7,8]; then, we specifically focus on the intentions of the agents. In this work, we define an intention as a plan to cover a specific landmark. We believe this is aligned with the common definition, where desires represent the goals of the agents and intentions represent concrete plans to achieve those goals [9]. Finally, commonly these works focus on the explicit prediction of beliefs [8]—i.e., can a deep neural network represent the beliefs of the other agent? In contrast, our work shows that the representations of intentions of other agents emerge as side-effects when training the agents with the task-specific objective function. This is called implicit modeling, and it is similar to the approach used in [10–12].

The most similar approach to ours has been previously applied in the context of perspective taking [12]. Compared to our work, they focused on a single-agent use case and modeling the field-of-view of another (static) agent. We found a similar result for modeling intentions in a cooperative multi-agent task. In the context of self-driving cars, the behavior prediction of other traffic agents is also a widely studied problem [3,4], but again, usually explicit prediction of future actions of other agents is used instead of implicit modeling of their intentions. The use of linear probes to understand intermediate layers of neural networks was proposed in [13].

Co-adaptation of trained agents to their training partners has been observed before in both cooperative and competitive tasks [14,15]. The proposed solutions [16,17] often assume that there are some similarities between the partners seen during training and new partners seen at test time, or that behaviors of other agents come from a fixed set. In our work, we show that exposure to a variety of behaviors at training time indeed benefits generalization. We also show that there are subtle ways in which generalization can be hindered, e.g., by having agents in a fixed order in the observation data structure.

## 3. Background

### 3.1. Multi-Agent Reinforcement Learning

A reinforcement learning problem is often modeled as a Markov decision process (MDP) [18]. In an MDP, an agent acts in an environment that has a state $s \in \mathcal{S}$, and the agent uses it to choose its action $a \in \mathcal{A}$ according to its stochastic policy $\pi(a|s)$. Taking this action produces scalar reward $r = R(s, a)$ and causes the environment to change its state according to probability distribution $P(s'|s, a)$. The initial state $s_0$ is chosen from some distribution $\sigma$.

The process of the agent choosing an action and environment transitioning to a new state is repeated until the end of the episode, and it produces a trajectory of states, actions and rewards $\tau = \{s_0, a_0, r_1, s_1, a_1, r_2, s_2, \ldots, r_T, s_T\}$, where $T$ is the length of the trajectory. The agent aims to find an optimal policy $\pi^*$ that maximizes its total expected return $R(\tau) = \sum_{t=0}^{T} \gamma^t r_t$, where $\gamma$ is the discount factor to make future rewards less valuable than immediate rewards.

In a partially observed MDP (POMDP) [19], the agent does not see the full environment state, just a partial observation $o = \omega(s)$. Therefore, the agent chooses its actions according to a policy conditioned on observations, not on states: $\pi(a|o)$.

The extension of POMDP to a multi-agent case is called a Markov game [20]. It largely follows the same formalism, except that each agent $i$ sees its own observation $o^i = \omega^i(s)$ and has its own policy $\pi^i(a^i|o^i)$ and reward function $r^i = R^i(s, a^i)$. When transitioning to a new state, the state transition function takes into account simultaneous actions from all $N$ agents: $P(s'|s, a^1, \ldots, a^N)$. The trajectory has the same form: $\tau = \{s_0, \mathbf{a}_0, \mathbf{r}_1, s_1, \mathbf{a}_1, \mathbf{r}_2, s_2, \ldots\}$, where $\mathbf{a}_t$ and $\mathbf{r}_t$ represent vectors of actions and rewards from all agents. Each agent $i$ aims to maximize its own total expected return: $R^i(\tau) = \sum_{t=0}^{T} \gamma^t r_t^i$.

### 3.2. Value Functions

Many algorithms for reinforcement learning make use of value functions that represent the "goodness" of a state or an action in terms of potential future rewards [21]. A state-value function represents the total expected return from a state $s$ using policy $\pi$:

$$V^{\pi}(s) = \mathop{\mathbb{E}}_{\tau \sim \pi}[R(\tau)|s_0 = s],$$

where expectation is established for the trajectories induced by policy $\pi$. Similarly, an action–value function or Q-function represents the total expected return when choosing action $a$ in state $s$:

$$Q^{\pi}(s,a) = \mathop{\mathbb{E}}_{\tau \sim \pi}[R(\tau)|s_0 = s, a_0 = a].$$

The optimal action–value function represents the maximum return by the best possible policy:

$$Q^*(s,a) = \max_{\pi} Q^{\pi}(s,a).$$

It can be found through temporal-difference learning [21,22], which means iteratively updating Q-function according to Bellman equation:

$$Q^*(s,a) = \mathop{\mathbb{E}}_{s' \sim P}\left[R(s,a) + \gamma \max_{a'} Q^*(s',a')\right].$$

Once the optimal action–value function is found, it can be turned into an optimal policy by just finding the highest-scoring action for a given state:

$$\pi^*(s) = \operatorname*{argmax}_{a} Q^*(s,a).$$

### 3.3. Deep Deterministic Policy Gradient (DDPG)

Policy gradient methods for reinforcement learning [23] adjust the parameters $\theta$ of policy $\pi_\theta$ directly to maximize the objective

$$J(\theta) = \mathop{\mathbb{E}}_{\tau \sim \pi_\theta}[R(\tau)].$$

Policy gradient theorem [23] states that the gradient of the above objective can be written as

$$\nabla_\theta J(\theta) = \mathop{\mathbb{E}}_{\tau \sim \pi_\theta}\left[\sum_{t=0}^{T}(\nabla_\theta \log \pi_\theta(a_t|s_t))Q^{\pi}(s_t,a_t)\right].$$

The state–value function $Q^{\pi}$ is called a critic in this context, and policy $\pi_\theta$ is called an actor. The algorithms using this form of policy update are called actor–critic algorithms [21].

Deterministic policy gradient (DPG) [24] allows learning deterministic policies for continuous action spaces by plugging the deterministic policy function $a = \mu_\theta(s)$ directly into the objective function of expected action–value $Q^{\mu}_\phi$ for all states:

$$J(\theta) = \mathop{\mathbb{E}}_{s \in \mathcal{S}}\left[Q^{\mu}_\phi(s, \mu_\theta(s))\right].$$

In this case, the gradient expression for policy weights $\theta$ has the following form:

$$\nabla_\theta J(\theta) = \mathop{\mathbb{E}}_{s \in \mathcal{S}}\left[\nabla_\theta \mu_\theta(s) \nabla_a Q^{\mu}_\phi(s,a)|_{a=\mu_\theta(s)}\right].$$

The algorithm alternates between learning Q-function $Q_\phi^\mu$ and policy function $\pi_\theta$ using the above equation. Q-function parameters $\phi$ are learned using temporal-difference learning by minimizing the mean-squared Bellman error over collected dataset $\mathcal{D}$:

$$L(\phi) = \mathop{\mathbb{E}}_{(s,a,r,s')\sim\mathcal{D}}\left[\left(Q_\phi^\mu(s,a) - (r + \gamma Q_\phi^\mu(s', \mu_\theta(s')))\right)^2\right].$$

Deep deterministic policy gradient (DDPG) [25] approximates Q-function $Q_\phi^\mu$ and policy $\mu_\theta$ using deep neural networks and makes use of replay memory and target networks to stabilize the learning [26].

### 3.4. Multi-Agent Deep Deterministic Policy Gradient (MADDPG)

The multi-agent deep deterministic policy gradient (MADDPG) algorithm [5] adapts the DDPG algorithm to multi-agent settings. It can be used with both cooperative and competitive tasks, and it does not make any assumptions regarding the environment or communication channel between the agents (e.g., differentiability). It adopts the framework of centralized training and decentralized execution, which means that information about all agents is used at training time to learn the action–value function (critic), but the policy function (actor) relies only on individual agent observation at execution time (see Figure 2).

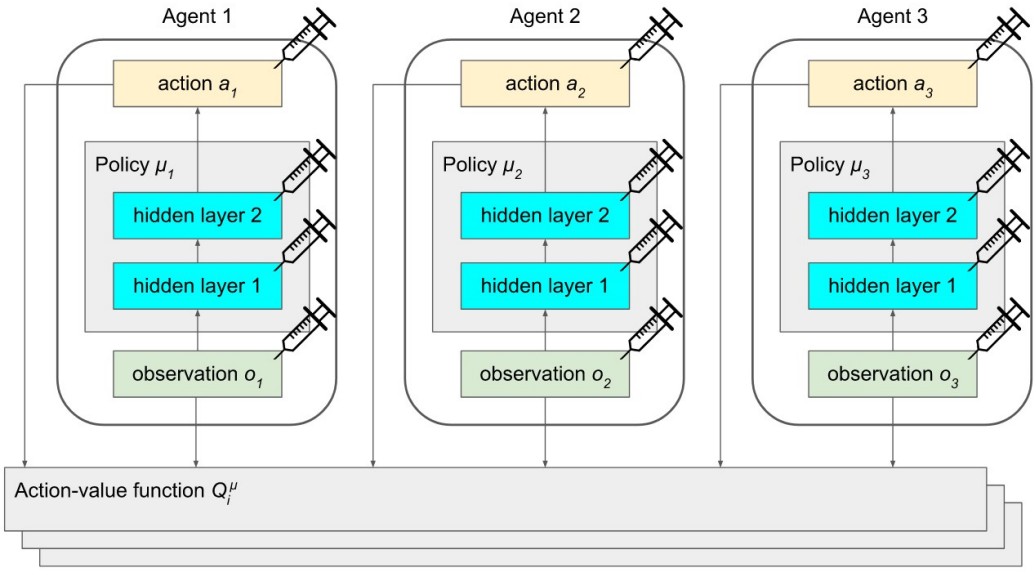

**Figure 2.** The centralized training decentralized execution framework proposed in the multi-agent deep deterministic policy gradien (MADDPG) algorithm. The action–value function $Q_i^\mu$ for agent $i$ makes use of observations and actions of all agents but is used only at training time. During the evaluation, the policy function $\mu_i$ for each agent $i$ depends only on the agent's local observation $o_i$ when producing action $a_i$. In this work linear probes depicted by syringe are used at observation, hidden layer 1, hidden layer 2, and action level to predict the final landmark of other agents.

In particular, given observations $\mathbf{x} = (o_1, \ldots, o_N)$, actions $\mathbf{a} = (a_1, \ldots, a_N)$, policies $\boldsymbol{\mu} = (\mu_1, \ldots, \mu_N)$ parameterized by $(\theta_1, \ldots, \theta_N)$ and action–value functions $(Q_1^{\boldsymbol{\mu}}, \ldots, Q_N^{\boldsymbol{\mu}})$ parameterized by $(\phi_1, \ldots, \phi_N)$, the policy for agent $i$ is learned using following gradient update:

$$\nabla_{\theta_i} J(\theta_i) = \mathop{\mathbb{E}}_{\mathbf{x},\mathbf{a}\sim\mathcal{D}}\left[\nabla_{\theta_i}\mu_i(a_i|o_i)\nabla_{a_i}Q_i^{\boldsymbol{\mu}}(\mathbf{x}, a_1, \ldots, a_N)|_{a_i=\mu_i(o_i)}\right]$$

The key difference from vanilla DDPG gradient expression is that observations and actions of all agents are used to estimate the action–value and are assumed to be accessible at training time. At the same time, each agent has its own action–value function, which allows them to have differing objectives, e.g., competition. For simplicity of notation,

the state is shown as consisting of the observations of all agents, but it can contain also additional information. The critic is learned by minimizing the following loss function:

$$L(\phi_i) = \mathop{\mathbb{E}}_{\mathbf{x},\mathbf{a},\mathbf{r},\mathbf{x}'\sim\mathcal{D}}\left[\left(Q_i^{\boldsymbol{\mu}}(\mathbf{x},a_1,\dots,a_N) - y\right)^2\right], y = r_i + \gamma Q_i^{\boldsymbol{\mu}'}(\mathbf{x}',a_1',\dots,a_N')\big|_{a_i'=\mu_i'(o_j')},$$

where $\boldsymbol{\mu}' = (\mu_1',\dots,\mu_N')$ is the set of target policies with delayed parameters $(\theta_1',\dots,\theta_N')$.

*3.5. Cooperative Navigation Task*

Our experiments are based on a cooperative navigation task described in [5]. In this task, three agents try to cover three landmarks and have to coordinate which agent covers which landmark (see Figure 1). The reward at every time step is

$$r = -\sum_i min_j(d_{ij}) - c$$

where $d_{ij}$ is the distance from landmark $i$ to agent $j$ and $c$ is the number of collisions. These rewards incentivize each landmark to have exactly one agent close to it and to have as few collisions as possible. The observation of each agent consists of 14 real values: the velocity of the agent (2), the position of the agent (2), the egocentric coordinates of all landmarks (6), and the egocentric coordinates of all other agents (4). The action of an agent consists of five real values: acceleration in four possible directions (only positive values) and a dummy value for no action. Accelerations in opposing directions are summed up.

## 4. Methods

We followed the training scheme from [5] and used the MADDPG algorithm with default settings. The network had two fully connected hidden layers with 128 nodes, each followed by rectified linear unit (ReLU) non-linearity. We trained the agents for 100,000 episodes, which was approximately when the convergence happened. Thereafter, we evaluated the frozen agents for 4000 episodes and recorded the observations, hidden-layer activations (from both layers) and actions of each agent.

To decode other agents' intentions, we trained a linear readout probe to predict the indexes of the final landmarks (the landmark covered at the final timestep of the episode) of other agents from each agent's hidden-layer activations (see Figure 3 left). This is treated as a classification task with three landmarks as the three classes (the readout model has three outputs). We used 4-fold cross-validation over 4000 episodes and report the mean classification accuracy over folds. The training was done separately for each of the 25 timesteps (all episodes were of the same length). Only those episodes where each other agent covered just one landmark were considered. For comparison, we also trained readout models to predict the final landmark from the observation (network's input) and the actions (network's output). The use of a linear decoder model guarantees that the information must be present explicitly in the decoder model's input.

To test the generalization ability of the agents, we put them together with two "Sheldon" agents—agents that each go straight to a fixed landmark ("their spots"). This leaves one landmark free for the trained agent to cover (see Figure 3 right). We ran an evaluation for 9 possible combinations of 3 co-trained agents and 3 free landmarks. One evaluation lasted 4000 episodes, and we report the percentage of episodes where all landmarks were covered. We compare this with the same measure from previous evaluations where all agents were co-trained agents. All results were averaged over five random training seeds.

We tried modifications to the MADDPG algorithm to alleviate the generalization issues:

- **MADDPG + shuffle:** Randomize the order of agents for each episode. The order determines the position of other agents' data in an agent's observation. Randomizing makes it impossible to have fixed assumptions about other agents' behavior.
- **MADDPG + shared:** A shared model for all agents is used, making them basically equivalent and eliminating the option for landmark preferences.

- **MADDPG + ensemble:** An ensemble of agents for each position is used, as suggested in [5]. The agents in an ensemble develop different policies because of different random initialization and different training samples they see from replay memory. This increases the diversity of partners any agent experiences, which forces it to develop more general strategies.

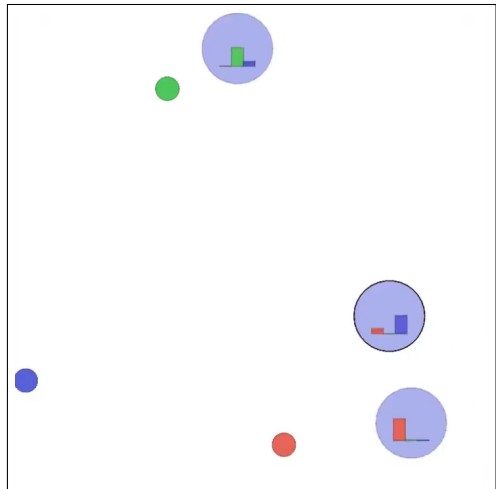 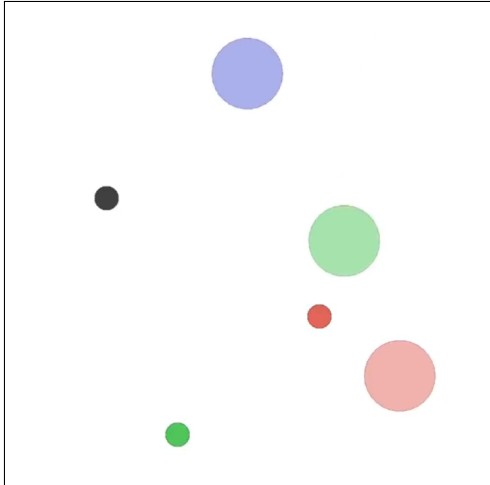

**Figure 3.** The tasks in this work. (**Left**): Reading out intentions of other agents. The hidden layer of the agent with the black circle is used to predict the final target of all agents, including itself. The bars inside each agent show probabilities of this agent going to the landmark of the same color. (**Right**): Generalization with "Sheldon" agents. Red and green agents always go to red and green landmarks. The trained agent in purple has to figure out it has to cover the only remaining black landmark.

## 5. Results

### 5.1. Reading Out Intentions

We observed that the agents jointly trained with the vanilla MADDPG had preferences for certain landmarks (see also Section 5.2), and hence, in many cases the readout probe could achieve good classification performance by just producing a constant prediction (see Figure A1 in Appendix A). To remove such bias, we used the MADDPG + shuffle scheme for assessing the decoding performance. Figure 4 shows the readout probe's prediction results.

We observe that the agent predicts its own final landmark generally better than others, which is expected because it has direct access to the hidden state that guides the actions.

The final landmarks of other agents can be predicted numerically better from hidden-layer-1 activations than from observations or from hidden-layer-2 activations. While all relevant information is already contained within the observation (because hidden-layer activations are computed from observations), it is more explicitly represented in the hidden layers and can be successfully decoded with a simple linear model. Representations in hidden layer 2 presumably focus more on the policy (the actions to be chosen), and therefore lose some of the information about the intention.

The output of the network (the actions) was uninformative for predicting the final landmark of other agents, which was expected. Prediction accuracy from observations and actions was close to chance (33%), but not exactly. This can be explained by there still being some landmark preferences in agents, which can be learned by the linear readout probe.

The accuracy of landmark prediction for other agents from hidden layer 1 ranged from 55% to 80% over time. While not perfectly accurate, it is clearly above the level of chance—33% (see Figure A2 in Appendix A for *p*-values). Interestingly, the final landmarks of other agents can be predicted better than chance even in the first timestep, possibly by assuming that each agent goes to the closest landmark. There was also an increase in prediction accuracy as the episode progressed, which was expected.

However, we note that even in the final steps of the episode, the accuracy of landmark prediction for other agents did not reach 100%. From observations, this can be explained by the linear nature of the decoder, which cannot represent the distance function needed to assess the closest landmark to each agent. From hidden-layer representations, we note that, while they would be capable of computing the closest agent to each landmark, this is presumably not needed to solve the task. All the agent needs is to infer if its intended landmark will be available, and not which other agent goes to which other landmarks.

The video showcasing the intention readouts is available here: https://youtu.be/3-pMUPPo970, (accessed on 26 December 2022).

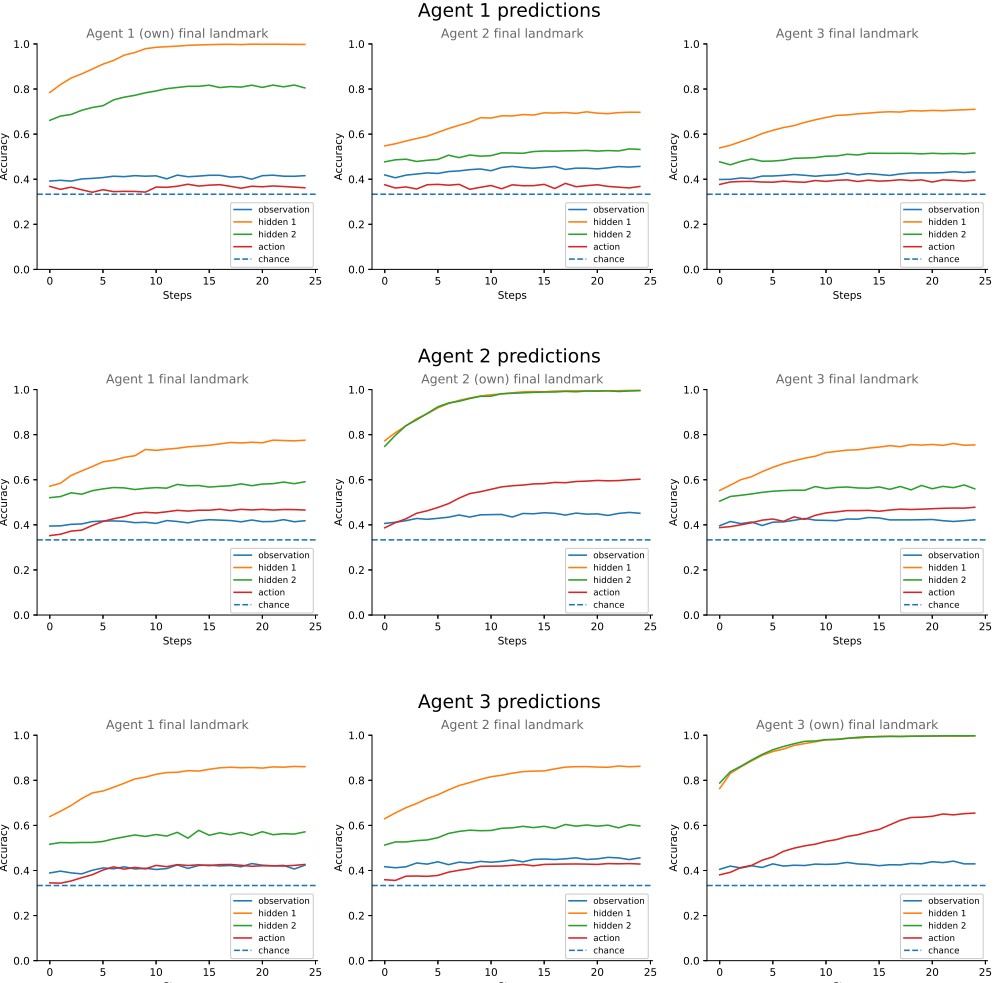

**Figure 4.** Cross-validated accuracy of a linear read-out probe for the MADDPG + shuffle scheme. All 9 combinations of 3 agents predicting the final landmarks of the other 3 agents are shown, including the agent predicting its own final target. The y-axis is the cross-validated accuracy of a linear read-out probe, and the x-axis is the timestep of an episode.

## 5.2. Generalization Gap

As noted before, agents jointly trained with the vanilla MADDPG algorithm developed preferences for certain landmarks (see Table 1). In addition to making reading out intentions uninformative due to a biased dataset, it was also clear that the trained agents had co-adapted to each other, and as such would not generalize when put together with agents unseen during training.

**Table 1.** Percentage of episodes where a given agent covered a given landmark with the vanilla MADDPG algorithm. Only those episodes where all three landmarks were covered by exactly one agent (48% of all episodes) were counted. Agent and landmark numbers refer to their positions in observation.

|         | Landmark 1 | Landmark 2 | Landmark 3 |
|---------|------------|------------|------------|
| agent 1 | 74%        | 25%        | 1%         |
| agent 2 | 25%        | 75%        | 0%         |
| agent 3 | 1%         | 0%         | 99%        |

To quantify the lack of generalization, we put a trained agent together with two "Sheldon" agents, each of which always goes to a fixed landmark. This test measures the ability of the agent to adapt its policy to an unforeseen situation where only one landmark is free for it to achieve the goal. Indeed, the agents showed an inability to adapt to the situation when the free landmark was their least favorite; see Table 2 for an example.

**Table 2.** Percentage of episodes where agents were able to cover all landmarks, when two "Sheldon" agents covered two fixed landmarks and the given agent trained with the vanilla MADDPG had only one free landmark to cover. Agent and landmark numbers refer to their positions in observation.

|         | Landmark 1 | Landmark 2 | Landmark 3 |
|---------|------------|------------|------------|
| agent 1 | 65%        | 42%        | 3%         |
| agent 2 | 28%        | 75%        | 2%         |
| agent 3 | 5%         | 2%         | 78%        |

Notice that these tables are not directly comparable—numbers in Table 1 represent the frequencies of choosing particular landmarks, but numbers in Table 2 represent the fractions of episodes where agents covered all three landmarks. Instead, the numbers in Table 2 should be compared with the reference number 48% achieved when the agent was paired with two co-trained agents. When the target is the least preferred, then the performance drops, but when the target is the most preferred, then the performance actually improves, as the "Sheldon" agents are very reliable partners. Though Tables 1 and 2 showcase a single training run for clarity, a similar pattern occurred in all runs.

*5.3. Improving Generalization*

To improve the generalization of the MADDPG algorithm, we tried out three different modifications, as described in Section 4. To evaluate generalization, we compare the performance of an agent against co-trained agents with the performance of the same agent against "Sheldon" agents.

We report the percentage of evaluation episodes where the agents covered all three landmarks. This number was averaged over five training runs, and in the case of generalization experiments also over nine agent-landmark combinations, similarly to Table 2. Figure 5 shows the results.

We make the following observations:

- Vanilla MADDPG agents achieved very good results when evaluated against other co-trained agents, but failed when confronted with agents with unseen policies. A large standard deviation with "Sheldon" agents resulted from the fact that for favorite landmarks, the success rate was very high, and for the least favorite landmarks, the success rate was very low (see Table 2 for an example).
- The MADDPG + shared scheme was much worse than vanilla MADDPG when evaluated against co-trained agents, but surprisingly achieved better generalization against "Sheldon" agents. This can be explained by the fact that "Sheldon" agents always succeed in doing their part, whereas co-trained agents often fail due to similar preferences.

- The MADDPG + shuffle scheme achieved a consistent success rate both with co-trained agents and with "Sheldon" agents. Though improving with "Sheldon" agents, the performance with co-trained agents was worse compared to the vanilla MADDPG.
- The MADDPG + ensemble scheme improved the result with co-trained agents but did not fix the generalization gap; the ensembles still developed favorite landmarks.

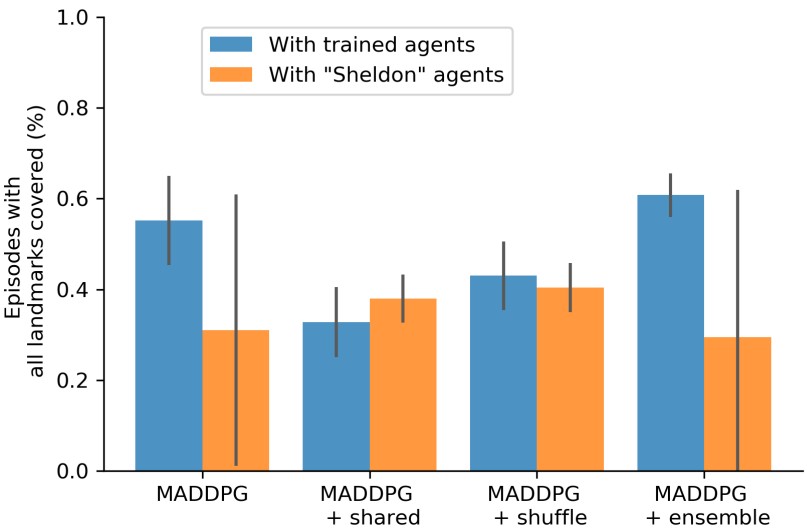

**Figure 5.** The percentage of episodes where all three landmarks were covered for each training scheme. The blue bars represent the means and standard deviations over 5 runs for all trained agents. The orange bars represent the means and standard deviations over $5 \times 9$ runs with two "Sheldon" agents and one trained agent. For numerical results and significance *p*-values, see Tables A1 and A2 in Appendix A.

We note that in only about half of the episodes, all three landmarks were covered by the agents. The best result with co-trained agents was 60.8%, and the best result with "Sheldon" agents was 40.4%. Usually, the reason was that one of the three agents failed to cover its landmark, rendering the whole episode a failure. In a small number of cases, the landmarks were generated very close to each other, which made it impossible to cover them with different agents. The behavior of agents matched the behavior in the original work [5]; see Table A3.

The video showcasing the generalization improvements is available here: https://youtu.be/r5jMpdC_pSk, (accessed on 26 December 2022).

## 6. Discussion

### 6.1. Reading Out Intentions

Inferring the intentions of others is the basis for effective cooperation [27]. We studied a simplified version of cooperation where three agents needed to cover three landmarks. We observed that the agents learn to model the intentions of other agents: even at the beginning of the episode, the activity of the hidden layers of a particular agent carried information about the other agents' plans. While the experiment was done with the MADDPG algorithm, we do not think the results are much dependent on the training methodology, but rather on the representation of the policy, which was an artificial neural network in this case. We would also like to point out that the judgment of other agents' intentions was made based on a single observation, consisting of the positions of other agents and locations of landmarks. While this makes our intention reading rather simplistic, we would contend that this aspect of intention modeling is a necessary building block for reading more complex intentions.

Studying human-level intentions (and ToM) has proven to be complicated [1,28–30]; studying simple tasks with agents whose representations can be examined will provide

unique insights into the emergence of more complex aspects of ToM [2]. In particular, being able to manipulate the network architecture and the components of the system allows one to answer which aspects really matter for solving a particular task. For example, inferring and generalizing intentions seems to require explicit memory (i.e., knowledge about the behavioral patterns of specific other agents), but our current work shows that rudimentary intention reading can be done even without a specific memory store, using a single observation. It might be possible that for more complex scenarios, recurrent neural networks or networks with external memory are needed. Another aspect that we consider interesting for future exploration is the performance of intention reading in different cooperative and competitive tasks while agents are equipped with communication channels.

*6.2. Generalization in Multiagent Setups*

The lack of generalization in reinforcement learning has been criticized before [31]. Especially problematic is the fact that we tend to test our agents in the same environment that they were trained in. Multiagent training adds another dimension to the generalization problem—the agents should also perform well against opponents and with partners unseen during training.

In the present work, we observed that when trained with the same partners, agents overfit to the behavior of their partners and cannot cooperate with a novel agent. While the issue of co-adaptation during training was also raised in [5], they mainly pointed it out in the context of competitive tasks. We show that it is just as much of a problem in cooperative tasks. Furthermore, we demonstrated that the ensembling approach suggested in [5] does not fix it. A more thorough analysis of the overfitting problem is presented in [15].

In principle, the solution is simple—the agents need to experience a variety of partners during training to generalize well. Similarly to data augmentation used in supervised training, we would need to "augment" our policies in various ways to produce the widest variety of training partners. Unfortunately, it is not clear how to achieve this in an automated and generalizable way. Proposed approaches range from learning maximum entropy policies [32], using quality-diversity algorithms [33], to taking inspiration from self-play [34–36]. However, it is not clear if that process would ever produce "Sheldon" policies used in our experiments, nor if they are actually needed.

For the evaluation of generalization, we chose agents that go to fixed landmarks. This was done for the simplicity and reproducibility of the experiments. At the same time, we do not think it is limiting, because by evaluating all possible combinations of each agent against two reserved landmarks we practically test all reasonable partner behaviors. In particular, this is equivalent to having two other agents randomly pick two landmarks. Having two fixed partners always succeed also guarantees that we test the trained agent, not our heuristic for the other agents.

People have experienced instabilities in self-play in competitive tasks [36,37]. We report similar instabilities in self-play in cooperative tasks. Our proposed MADDPG + shared algorithm sometimes performed surprisingly well with partners not seen during training, but other times failed to learn good policies. Still, it holds the promise of possibly learning more general policies than other approaches.

The current work did not consider deception, as in cooperative multi-agent tasks, there is no incentive to deceive other agents. However, deception, in particular, discriminating a real intention from the fake one is an important property of competitive multi-agent tasks. We leave it as future work and refer to [38] for further reading.

The ability of agents to make decisions having only limited information about the environment and within limited computational power is called bounded rationality [39]. Partial observability in multi-agent Markov games is directly related to the concept of bounded rationality, as the full knowledge of the environmental state or the states of other agents might not be known. Although our task happens to be fully observable, the underlying MADDPG algorithm has been shown to work also in partially observable environments [5]. Therefore, we believe that the ability to model intentions depends more

on the representation of the policy (as a neural network) than the observability of the environment. We also note that the feed-forward network used in our agents uses fixed computational power, and therefore, basic intention modeling can be done already within the limits of bounded rationality.

## 7. Conclusions

When the agent controlling a self-driving car performs an unprotected left turn at an intersection, it needs to cope with any kind of behavior from other drivers, however hostile or incompetent they are. Inferring the intentions of other agents is therefore crucial to behaving in a reliable manner. For example, the controlling agent of a self-driving car cannot expect all other cars to run the same version of the same software, or even any software at all. Thus, they need to generalize to unforeseen situations and behaviors of other drivers.

In this work, we showed that deep reinforcement learning agents trained with the MADDPG algorithm indeed learn to model the intentions of other agents when trying to solve a cooperative task. In a cooperative navigation task, the final target of another agent could be predicted better from the hidden-layer activations than from the observation. As the hidden layer is computed from observation, the learned transformation applied by the agent must make this information more explicit so that it can be read out better with a linear decoder. This also confirms that linear read-out probes are a good technique for examining the learned network.

Trying to read out the intentions of agents exposed the lack of generalization in learned models. Trained agents co-adapted to specific behaviors of each other and failed consistently when put together with unseen agents. We showed that simple shuffling of all agents at each episode improves the generalization, whereas ensembling does not. While this alleviates the generalization gap somewhat, a more robust solution would be to train the agents against opponents using a diverse range of policies.

The proposed method for confirming intention modeling in deep learning agents is simple to implement and can be used to improve the generalization of multi-agent systems in fields such as robotics, autonomous vehicles, and smart cities.

**Author Contributions:** Conceptualization, T.M., J.A. and R.V.; funding acquisition, R.V.; investigation, A.L. and D.M.; methodology, T.M.; supervision, J.A. and R.V.; visualization, A.L. and D.M.; writing—original draft, T.M.; writing—review and editing, J.A. and R.V. All authors have read and agreed to the published version of the manuscript.

**Funding:** The authors thank the financial support from The Estonian Research Council through the personal research grant PUT1476. We also acknowledge funding by the European Regional Development Fund through the Estonian Center of Excellence in IT, EXCITE, project number TK148. R.V. also thanks the Estonian Research Council grant PRG1604 and the European Union's Horizon 2020 Research and Innovation Programme under grant agreement number 952060 (Trust AI). J.A. also thanks the Estonian Research Council grant PSG728 and European Social Fund via IT Academy programme.

**Data Availability Statement:** The code is available at https://github.com/NeuroCSUT/intentions, (accessed on 26 December 2022).

**Acknowledgments:** T.M. would like to thank Ryan Lowe, Yi Wu and Ardi Tampuu for helpful discussions.

**Conflicts of Interest:** The authors declare no conflict of interest.

## Appendix A

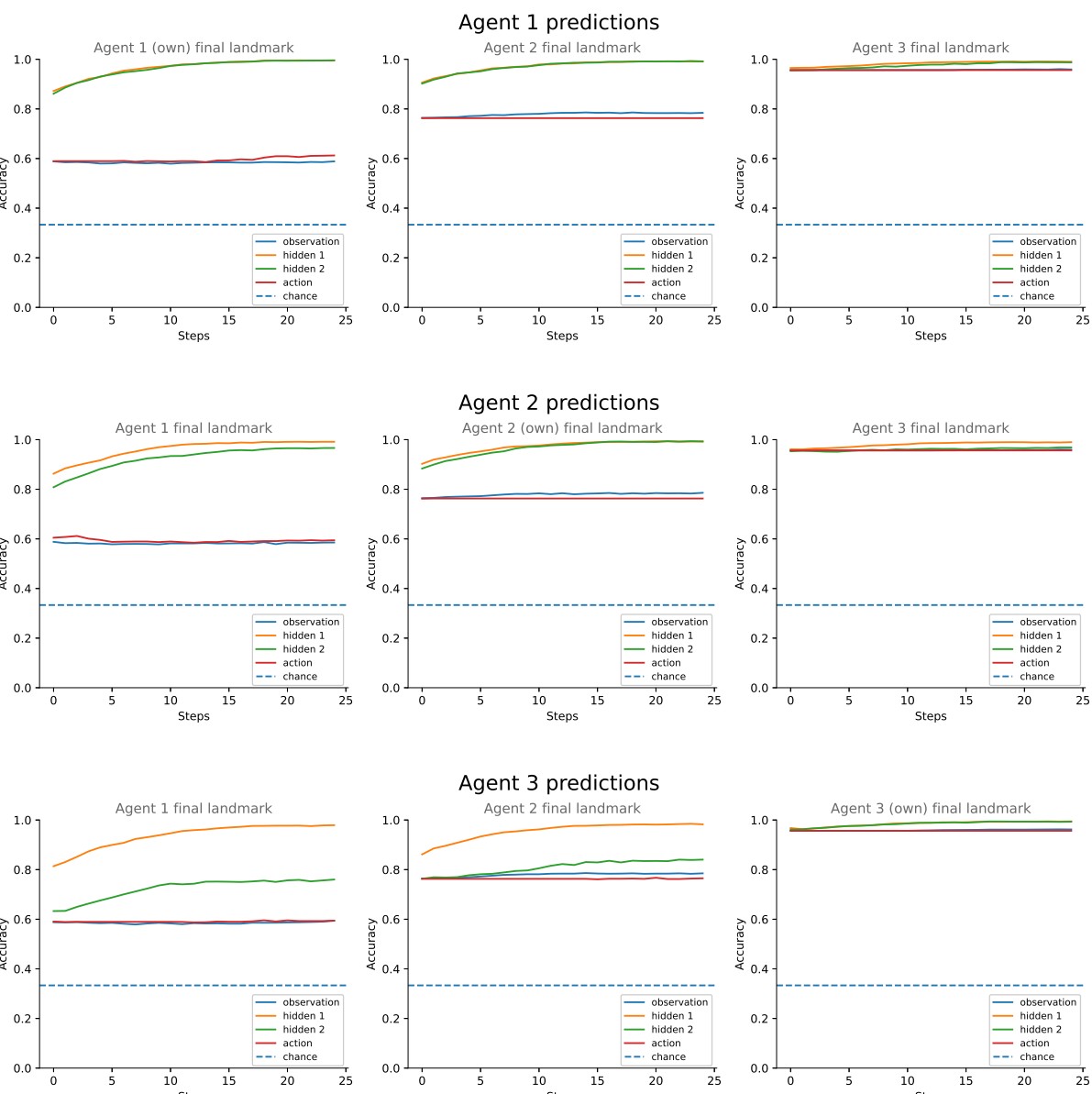

**Figure A1.** Cross-validated accuracy of linear read-out probe for the vanilla MADDPG algorithm. The predictions are unusually good because of biases of learned agents—the read-out model can score well by just producing constant prediction. For example, agent 3 can be predicted perfectly because in 99% of the cases, it goes to landmark 3 (see Table 1). Still, the comments from Section 5.1 hold.

**Table A1.** Numerical results shown in Figure 5. Success rate refers to the fraction of episodes where agents covered all three landmarks. The best result for each block (up to significance shown in Table A2) is shown in bold. Notice that the results consist of 5 values in the case of co-trained agents and 45 values in the case of "Sheldon" agents. Every single value by itself is the fraction of episodes where all agents covered all landmarks out of 4000 evaluation episodes.

| | Method | Success Rate ± Std. |
|---|---|---|
| With co-trained agents | MADDPG | **0.552 ± 0.098** |
| | MADDPG + shared | 0.328 ± 0.077 |
| | MADDPG + shuffle | 0.43 ± 0.076 |
| | MADDPG + ensemble | **0.608 ± 0.048** |
| With "Sheldon" agents | MADDPG | 0.31 ± 0.299 |
| | MADDPG + shared | 0.38 ± 0.053 |
| | MADDPG + shuffle | **0.404 ± 0.055** |
| | MADDPG + ensemble | 0.295 ± 0.324 |

**Table A2.** Significance ($p$-values) for means in Figure 5 calculated using a two-sided permutation test. Cases where the means are significantly different ($p$-value < 0.05) are marked as bold. Notice that the results consist of 5 values in the case of co-trained agents and 45 in the case of "Sheldon" agents. Every single value was the fraction of episodes where all agents covered all landmarks out of 4000 evaluation episodes.

| | Method A | Method B | $p$-Value |
|---|---|---|---|
| With co-trained agents | MADDPG | MADDPG + shared | **0.016** |
| | MADDPG | MADDPG + shuffle | 0.087 |
| | MADDPG | MADDPG + ensemble | 0.333 |
| | MADDPG + shared | MADDPG + shuffle | 0.087 |
| | MADDPG + shared | MADDPG + ensemble | **0.008** |
| | MADDPG + shuffle | MADDPG + ensemble | **0.008** |
| With "Sheldon" agents | MADDPG | MADDPG + shared | 0.131 |
| | MADDPG | MADDPG + shuffle | **0.044** |
| | MADDPG | MADDPG + ensemble | 0.817 |
| | MADDPG + shared | MADDPG + shuffle | **0.036** |
| | MADDPG + shared | MADDPG + ensemble | 0.089 |
| | MADDPG + shuffle | MADDPG + ensemble | **0.030** |

**Table A3.** Metrics reported in [5] for experiments. Average distance refers to the distance from each landmark to the closest agent, averaged over episodes, timesteps, and landmarks. Averaging over timesteps is used to take into account how fast the agents achieve the final target. # collisions refers to the average number of collisions over episodes and timesteps. We also show results from [5] for context, but it was impossible to exactly match them, as the codebase had changed since publication. It is also possible that there was some mistake in the original publication; e.g., average distance and # collisions could have been switched, or average distance could have actually been the sum of closest agent distances averaged over episodes and timesteps. We tried to clarify this with the original authors, who suggested running the latest version of the published code and using it as a baseline. This is the "MADDPG with co-trained agents" line.

| | Method | Average Dist. | # Collisions |
|---|---|---|---|
| With co-trained agents | MADDPG | 0.221 ± 0.006 | 0.500 ± 0.043 |
| | MADDPG + shared | 0.220 ± 0.010 | 0.304 ± 0.079 |
| | MADDPG + shuffle | 0.215 ± 0.009 | 0.488 ± 0.075 |
| | MADDPG + ensemble | 0.212 ± 0.007 | 0.532 ± 0.035 |
| With "Sheldon" agents | MADDPG | 0.299 ± 0.049 | 6.289 ± 2.815 |
| | MADDPG + shared | 0.280 ± 0.004 | 4.772 ± 0.342 |
| | MADDPG + shuffle | 0.283 ± 0.007 | 4.933 ± 0.739 |
| | MADDPG + ensemble | 0.303 ± 0.052 | 6.829 ± 3.043 |
| Lowe et al. [5] | MADDPG | 1.767 | 0.209 |

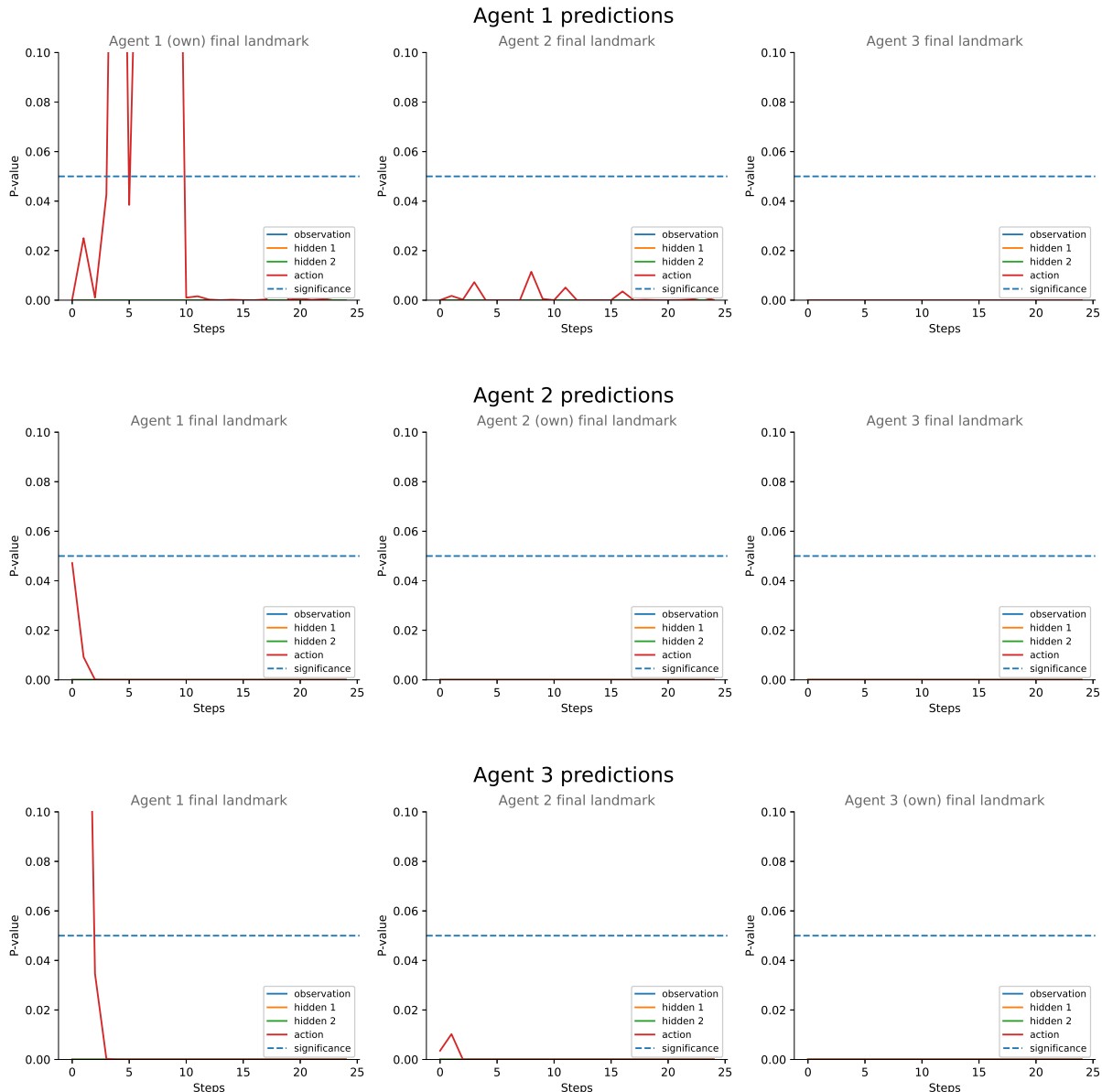

**Figure A2.** Significance for the accuracies shown in Figure 4. The *p*-values were calculated using binomial test against chance probability 33%. This shows most of the results have very low *p*-values, and therefore, were significant, with the exception of predicting from action, which was very close to random chance, as expected. Non-random prediction from observation and action can be explained by some remaining bias in the agents to prefer certain landmark, which can be learned by the linear readout probe.

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
