# Peer review of "Do Deep Reinforcement Learning Agents Model Intentions?†"

_stats, doi:10.3390/stats6010004_

Round 1

Author Response

We would like to thank the reviewer for pointing out some mistakes and unclarities in the manuscript. We have since then updated the manuscript and here are to point-by-point responses:

  1. We thank the reviewer for pointing out the missing reference. We added references to Figure 3 to the manuscript.
  2. We thank the reviewer for pointing out unclear wording in the text. By hidden states we actually meant hidden layer activations, which are calculated from a 14-element input vector by multiplying it with a 14x128 learned weight matrix, adding a 128-element learned bias vector and passing it through element-wise rectified linear unit non-linearity, as usual with neural networks. The second layer activations are calculated similarly by multiplying the first layer activations with 128x128 learned weight matrix, adding a 128-element learned bias vector and passing it through element-wise rectified linear unit non-linearity. Intentions are decoded from hidden layer activations using linear decoder, which basically means multiplying the 128-element hidden vector with 128x3 learned weight matrix, adding 3-element learned bias vector and passing the result through softmax activation function that produces probabilities for three landmarks. These probabilities are interpreted as intentions of the agent, which landmark it plans to cover. We clarified it in the relevant sections.
  3. We thank the reviewer for pointing out unclear wording in the text. In fact we trained different linear readout models for observations, actions and hidden layers. These readout models are trained to predict the final landmark covered by other agents. These models can have different input, but are trained on basically the same data and share the same ground truth. This allows us to compare how the intention decoding improves or degrades throughout the agent processing pipeline from observations to hidden layers to actions. The results are shown in Figure 4. We clarified it in the relevant sections.
  4. We thank the reviewer for the question. The intention in this paper refers to the index of the landmark the agent plans to cover. Decoding the landmark location (as x and y coordinates) is a valid suggestion, but because the landmark location is already in the observation, we assumed it to be an easier task than decoding the landmark index. The use of the index of the landmark as intention was clarified in the manuscript.
  5. We thank the reviewer for the question. Reference [12] demonstrates that linear probes can be used to understand the representations in intermediate layers of a neural network. The key novelty of our work is that the linear probe technique can be used to validate intentions of agents represented as neural networks. While it is a widely held belief that neural networks are able to represent the intentions of other agents, to our best knowledge this paper is the first that empirically shows that they actually do that, as shown on Figure 4.

Reviewer 2 Report

The paper is super interesting and the results well-presented. I suggest to authors to make two remarks at the conclusion: 1) Deception: In reality, it is quite usual to spread intentionally false signals, aiming to deceive the other agents. In this case, how machine learning can discriminate the real intention from the fake one? 2) Bounded rationality: In reality, intelligent agents have neither perfect knowledge about the environment, nor the ability to process 100% effectively the received information from observations. This should be discussed. The authors may include the relevant paper in the references, making also a link with their work:  https://doi.org/10.3390/math9010103 

Author Response

We thank the reviewer for bringing important topics to our attention. We have since then updated the manuscript and here are the point-by-point responses:

  1. We thank the reviewer for bringing deception to our attention. We added the following passage to the end of Discussion section: “The current work does not consider deception, as in cooperative multi-agent tasks there is no incentive to deceive other agents. But deception, in particular discriminating the real intention from the fake one, is an important property of competitive multi-agent tasks. We leave it as a future work and refer to \cite{meta2022human} for further reading.”
  2. We thank the reviewer for bringing the concept of bounded rationality to our attention. We added the following passage to the Discussion section: “The ability of agents to make decisions having only limited information about the environment and within limited computational power is called bounded rationality \cite{gershman2015computational}. Partial observability in multi-agent Markov games is directly related to the concept of bounded rationality, as the full knowledge of the environment state or the state of other agents might not be known. Although our task happens to be fully observable, the underlying MADDPG algorithm has been shown to work also in partially observable environments \cite{lowe2017multi}. Therefore we believe that the ability to model intentions depends more on the representation of the policy (as a neural network) than the observability of the environment. We also note that the feed-forward network used in our agents uses fixed computational power and therefore basic intention modeling can be done already within the limits of bounded rationality.”

Round 2

Reviewer 1 Report

Thank you for modifications in the revised version. As I mentioned before, the claim that hidden layers of Neural Networks provides clear information about other agents intention is a very big claim, it needs more justifications, k-fold cross validation, statistical analysis or at least implementations in more complex scenarios. On the other hand the concept of intention in Multi agent system is different from what is alleviated in this manuscript being landmarks or goals in RL so it would probably be better to present the idea in a new wording.

Author Response

We thank the reviewer for the raised questions. We have updated text and figures in the manuscript, and also added one new figure in the appendix. Here are the point-by-point responses:

  1. We implemented k-fold cross-validation for estimating the accuracy of linear readout probes. This did not change the results, but made the graphs much smoother due to averaging over multiple folds. We have updated Figures 4 and A1 in the manuscript.
  2. We implemented significance testing for linear readout probe results. We used a binomial test against a random chance of 33%. Almost all results were strongly significant, with the exception of some predictions from actions, which was expected. The results are documented in Figure A2 in Appendix A.
  3. We clarified the terminology throughout the manuscript and added the following passage to the Related Work section: "In this work we define intention as a plan to cover a specific landmark. We believe this is aligned with the common definition, where desires represent the goals of the agents and intentions represent concrete plans to achieve those goals \cite{bratman1987intention}."

We hope that the above changes were helpful and addressed the comments to your satisfaction.